# Risk Factors for Recurrent Hip Fractures Following Surgical Treatment of Primary Osteoporotic Hip Fractures in Chinese Older Adults

**DOI:** 10.3390/diseases13110351

**Published:** 2025-10-27

**Authors:** Yuzhu Wang, Wenhui Shen, Jiayi Jiang, Lin Wang, Qing Xia, Yunchao Shao, Lu Cao

**Affiliations:** 1Department of Pharmacy, Zhongshan Hospital, Fudan University, Shanghai 200032, China; wang.yuzhu@zs-hospital.sh.cn; 2Department of Orthopedic Surgery, Zhongshan Hospital (Xiamen Branch), Fudan University, Xiamen 361000, China; xmswh@foxmail.com; 3Department of Nursing, Zhongshan Hospital, Fudan University, Shanghai 200032, China; jiang.jiayi@zs-hospital.sh.cn (J.J.); wanglin20667@126.com (L.W.); 4Department of Orthopedic Surgery, Zhongshan Hospital, Institute of Bone and Joint Diseases, Fudan University, Shanghai 200032, China; xia.qing@zs-hospital.sh.cn

**Keywords:** older adults, primary osteoporotic hip fracture, recurrent hip fracture, mortality, risk factors

## Abstract

Objectives: Hip fractures associated with osteoporosis are indicative of high rates of both disability and mortality. The objective of this study was to analyze the risk factors for recurrent hip fractures following primary osteoporotic hip fracture surgery in older adult patients. Methods: A single-center, retrospective cohort study was conducted on 376 patients suffering from primary osteoporotic hip fractures from 1 January 2020 to 31 December 2021. Multivariate logistic regression was used to identify risk factors for recurrent hip fractures. Results: The study observed 376 patients over a period of three years. The incidence of recurrent hip fractures was 20.5% (77/376). Multiple logistic regression analysis revealed that age ≥ 85 years (odd ratios [OR] = 3.127, 95% confidence interval [CI] = 1.672–5.849, *p* < 0.001), chronic obstructive pulmonary disease (COPD) (OR = 3.794, 95%CI = 1.747–8.236, *p* < 0.001), and Parkinson’s disease (PD) (OR = 2.744, 95%CI = 1.249–6.028, *p* = 0.012) were independent risk factors for recurrent hip fractures; antiosteoporosis drugs (OR = 0.243, 95%CI = 0.131–0.451, *p* < 0.001), duration of antiosteoporosis drug therapy (OR = 0.564, 95%CI = 0.283–0.830, *p* = 0.003) and serum albumin ≥ 35 g·L^−1^ (OR = 0.413, 95%CI = 0.194–0.881, *p* = 0.022) were independent protective factors for recurrent hip fractures. The receiver operating characteristic (ROC) curve demonstrated that the AUC was 0.802, the sensitivity was 77.8%, and the specificity was 75.5%. A significantly higher three-year mortality rate was observed among patients with recurrent hip fractures (26.0% vs. 15.4%, *p* = 0.029). Conclusions: Older patients with advanced age, COPD and PD were at greater risk of recurrent hip fractures. Early nutrition intervention and antiosteoporosis drug therapy may decrease the incidence of recurrent hip fractures in older patients, thereby reducing mortality.

## 1. Introduction

Osteoporosis (OP) is classified as a systemic bone disease. The condition is typified by reduced bone mass and impaired bone tissue microstructure, leading to elevated bone fragility [1]. As the global population continues to age, there is an increasing prevalence of osteoporosis and the subsequent occurrence of osteoporotic fractures. This situation gives rise to a marked menace with regard to public health [2,3,4,5]. The most severe complication associated with OP is an osteoporotic fracture, otherwise referred to as a fragility fracture. Such fractures may occur as a result of minor trauma or in the course of daily activities. Osteoporotic fractures have been shown to be associated with elevated rates of disability and mortality. These fractures most commonly manifest as those of the hip, the vertebrae and the wrist [6]. It is widely established that the presence of a primary fragility fracture is frequently indicative of osteoporosis and thus rendering the patient a risk factor for subsequent fractures [7].

Fragility fractures, particularly hip fractures in older individuals, have a deleterious effect on the quality of life of affected patients and their families. This is primarily attributable to the considerable financial constraints imposed on healthcare services, and the consequent loss of independence [6,8,9]. In accordance with the rising global life expectancy, it was anticipated that there would be a concomitant increase in the number of individuals who would sustain fragility fractures. Such fractures are regarded as a substantial disease burden for society, particularly in the Asian region [10]. While the definition of the osteoporosis burden as that determined by the occurrence of hip fractures or other fractures, or consequent disability, is valid, it may be more beneficial for the purpose of disease prevention to consider this burden in terms of those at high risk of a future fracture [11,12,13,14]. There is a broad consensus that surgical intervention for osteoporotic hip fractures is indicative of a more favorable prognosis. Nevertheless, the potential risk of subsequent hip fractures persists following surgical intervention, both ipsilateral and contralateral. Despite the fact that prior research has investigated the risk factors for refracture among older patients following hip fractures [15,16], studies on the risk of recurrent hip fractures following surgical treatment of primary osteoporotic hip fractures are relatively scarce, particularly in the Asian population. It is therefore of paramount importance to identify risk factors for recurrent hip fractures in older patients following surgical treatment of primary osteoporotic hip fractures, and to manage these at an early stage.

A substantial corpus of research has been dedicated to investigating the impact of treatment for a primary fracture on the occurrence of subsequent fractures and mortality [17,18,19,20,21]. Moreover, given the heterogeneity of national conditions across different countries, there is a possibility that the incidence and factors of fractures may vary between countries [22]. As demonstrated by clinical observations, it has been established that not all patients suffering from primary osteoporotic fractures will undergo subsequent hip fractures. Consequently, the present study hypothesizes that specific risk factors associated with recurrent hip fractures require dedicated consideration, and that a comprehensive and systematic investigation of risk factors for recurrent hip fractures is imperative. Moreover, existing prediction models may not fully incorporate neurological comorbidities among Chinese older patients, such as Parkinson’s disease and dementia. Patients suffering from Parkinson’s disease and dementia are predisposed to a heightened risk of falling, a phenomenon that can be attributed to impaired gait, postural regulation, and cognitive impairment [23]. Consequently, these factors may contribute to an elevated risk of fracture. This study was conducted with the objective of investigating risk factors and outcomes for recurrent hip fractures following primary osteoporotic hip fractures in the older Chinese population. The study’s findings may provide a theoretical foundation for effective prevention of recurrent hip fractures in older Chinese patients following surgical treatment of primary osteoporotic hip fractures.

## 2. Materials and Methods

### 2.1. Patient Enrollment and Eligibility Criteria

This retrospective cohort study was conducted at a medical center, namely Zhongshan Hospital, Fudan University. The present investigation was granted approval from the Medical Ethics Committee of Zhongshan Hospital, Fudan University (B2024-479R). The diagnosis of comorbidities in patients is based on the 10th Edition of the International Classification of Diseases (ICD-10). The data set under consideration comprises information on 376 patients diagnosed with primary osteoporotic hip fracture who underwent internal fixation or hip arthroplasty in the Department of Orthopaedic Surgery from 1 January 2020 to 31 December 2021. The following inclusion criteria were employed: (i) age ≥ 65 years; (ii) primary osteoporotic hip fractures caused by low-energy trauma; (iii) willingness to participate in the follow-up by telephone. The exclusion criteria were the following: (i) patients with multiple fractures, pathological fractures or open fractures; (ii) patients with osteoporosis induced by glucocorticoid therapy; (iii) patients who had immune system diseases, liver disease, renal insufficiency and tumor; (iv) patients with incomplete clinical data before and/or after surgery; (v) patients with recurrent vertebral fractures, wrist fractures and fractures of other site; (vi) unwillingness to participate in the study. In total, 376 out of 463 patients met the inclusion criteria and were subjected to retrospective follow-up over a period of three years following surgical treatment of primary osteoporotic hip fractures (Figure 1). A self-designed questionnaire was used, encompassing patients’ demographic information, comorbidities, whether antiosteoporosis medications were used, the duration of antiosteoporosis drug therapy, the time of recurrent hip fractures and the time of death.

The present study population comprised patients diagnosed with a hip fracture that met the criteria for osteoporosis. The present study utilized a three-year period of follow-up data to assess the incidence of recurrent hip fractures subsequent to primary osteoporotic fractures. A comprehensive review of the patients’ prescriptions and medications was conducted during the enrolment and follow-up procedures, with meticulous documentation being carried out accordingly. Follow-up time was defined as the number of days from the initial fracture event to the occurrence of recurrent hip fracture or the most recent appointment, whichever occurred first. The term “recurrent hip fracture” was defined as the time to the first occurrence after primary osteoporotic hip fracture surgery, with examples including ipsilateral or contralateral hip fractures. The diagnosis of hip fracture is made on the basis of the results of diagnostic imaging. The primary endpoint and statistical analysis were predetermined in advance. The primary endpoint of the study was defined as the timeframe to the initial occurrence of a subsequent hip fracture. The occurrence of mortality following surgical treatment of primary osteoporotic hip fractures was identified as the secondary endpoint.

### 2.2. Data Collection

The data was extracted from the hospital’s electronic medical records. The following variables were collected for the study: demographic characteristics (gender, age), body mass index (BMI), smoke, alcohol, ASA (American Society of Anesthesiologists, ASA) grade, site of primary hip fracture (left hip or right hip), surgical types of primary hip fracture (internal fixation or hip arthroplasty), types of primary hip fracture (femoral neck fracture or intertrochanteric fracture), anesthesia (spinal anesthesia or general anesthesia), blood transfusion, concomitant underlying diseases [hypertension, diabetes, heart attack, stroke, heart failure, chronic obstructive pulmonary disease (COPD), anxiety/depression, dementia, Parkinson’s disease (PD), deep vein thrombosis, and sleep disturbance], laboratory variables [estimated glomerular filtration rate (eGFR), serum albumin, hemoglobin (Hb), serum calcium, and 25-hydroxyvitamin D at admission]. Furthermore, data pertaining to the concomitant use of antiosteoporosis medications (bisphosphonates, denosumab, teriparatide) and economic factors was collected. The term “total costs” refers to the aggregate hospital costs associated with each hospitalization for a patient.

### 2.3. Statistical Analysis

All statistical analyses were conducted using the Statistical Package for the Social Sciences (SPSS) version 27.0 (IBM, 187 Chicago, IL, USA). The Kolmogorov–Smirnov test was used to determine the normality of the quantitative variables. Quantitative variables may be presented in two ways: as the mean and the standard deviation (SD), or as the median and the interquartile range (IQR). A comparison was conducted between the groups with respect to the variables, with either independent *t*-tests or rank-sum tests employed as appropriate. The Mann–Whitney *U* test was employed to compare the total costs incurred by the two groups. The analysis of qualitative variables was conducted by means of the chi-squared test or the Fisher exact test, expressing as frequencies and percentages. Furthermore, the present study evaluated the risk factors associated with recurrent hip fractures and mortality among older patients following surgical treatment of primary osteoporotic hip fractures by means of logistic regression models. Moreover, it was determined that certain variables necessitated the incorporation of forced entry into the multiple logistic regression models. A multiple logistic regression analysis was conducted on recurrent hip fractures and mortality, incorporating the following set of covariates: sex (female vs. male), age (65–84 years, ≥85 years), BMI (<19, ≥19, smoke (yes vs. no), alcohol (yes vs. no), ASA (grade 1, 2, 3), site of primary hip fracture (left or right), surgical types of primary hip fracture (internal fixation or hip arthroplasty), types of primary hip fracture (femoral neck fracture or intertrochanteric fracture), anesthesia (general anesthesia or spinal anesthesia), blood transfusion (yes vs. no), hypertension (yes vs. no), diabetes (yes vs. no), heart attack (yes vs. no), stroke (yes vs. no), heart failure (yes vs. no), COPD (yes vs. no), anxiety/depression (yes vs. no), dementia (yes vs. no), PD (yes vs. no), deep vein thrombosis (yes vs. no), sleep disturbance (yes vs. no), the use of antiosteoporosis medications (yes vs. no), eGFR, serum albumin, hemoglobin (Hb), serum calcium, and 25-hydroxyvitamin D. A backward conditional approach was adopted for the incorporation of novel terms into the logistics regression model. It is imperative to note that all *p* values were two-sided. In the statistical analysis conducted, a *p*-value of less than 0.05 was considered to be statistically significant. The subsequent development of the prediction model was predicated upon the prior identification of risk factors. Subsequently, the receiver operating characteristic (ROC) curve was employed to evaluate the model’s predictive capacity. All results were considered to be statistically significant at the *p* < 0.05 level (2-tailed).

## 3. Results

### 3.1. Incidence and Risk Factors for Recurrent Hip Fractures Following Surgical Treatment of Primary Osteoporotic Hip Fractures

The study comprised a total of 376 out of the 463 patients who met the inclusion criteria (Figure 1). The patients were subsequently divided into two groups: the non-recurrent hip fracture group (n = 299) and the recurrent hip fracture group (n = 77). The median time to recurrent hip fractures was 22.4 months, with an interquartile range (IQR) of 23.8 months. As illustrated by the data presented in Table 1, the patient population comprised 289 women and 87 men, thus yielding a female-to-male ratio of 76.9% to 23.1%, respectively. The age of the participants ranged from 65 to 98 years, with a mean age of 80.8 ± 8.3 years. The study population comprised 242 patients aged 65–84 years (64.4%), and 134 patients aged 85+ years (35.6%).

As shown in Table 1, the incidence of recurrent hip fractures in older patients following primary osteoporotic hip fracture surgery was 20.5% (77/376). A comparison of the two groups revealed a significant increase in the number of patients aged 85+ years in the recurrent hip fracture group, as opposed to the group without recurrent hip fractures (50.6% vs. 31.8%, *p* = 0.002). Recurrent hip fracture patients were more likely to have had alcohol consumption (26.0% vs. 12.4%, *p*= 0.015), and diagnosed with COPD (26.0% vs. 9.4%, *p* < 0.001), dementia (10.4% vs. 3.0%, *p* = 0.005), or PD (24.7% vs. 8.7%, *p* < 0.001). The utilization of antiosteoporosis drugs was found to be a more common occurrence in the cohort that demonstrated an absence of subsequent hip fractures (44.8% vs. 19.5%, *p* < 0.001). The median duration of antiosteoporosis drug therapy was found to be significantly longer in the non-recurrent hip fracture group than in the recurrent hip fracture group (36.0 vs. 33.9, *p* < 0.001). In relation to serum albumin levels of ≥35 g·L^−1^, a higher prevalence was demonstrated among patients in the non-recurrent hip fracture group compared to the recurrent hip fracture group (32.4% vs. 17.3%, *p*= 0.010).

Multiple logistic regression analysis revealed that age ≥ 85 years (odds ratio [OR] = 3.127, 95% confidence interval [CI] = 1.672–5.849, *p* < 0.001) was identified as an independent risk factor for recurrent hip fractures following primary osteoporotic hip fractures surgery; in addition, COPD (OR = 3.794, 95%CI = 1.747–8.236, *p* < 0.001) and PD (OR = 2.744, 95%CI = 1.249–6.028, *p* = 0.012) were also identified as independent risk factors for recurrent hip fractures. However, the use of antiosteoporosis drugs (OR = 0.243, 95%CI = 0.131–0.451, *p* < 0.001), the duration of antiosteoporosis drug therapy (OR = 0.564, 95%CI = 0.283–0.830, *p* = 0.003) and serum albumin levels of ≥35 g·L^−1^ (OR = 0.413, 95%CI = 0.194–0.881, *p* = 0.022) were determined to be independent protective factors for recurrent hip fractures following primary osteoporotic hip fractures surgery (Table 2).

### 3.2. ROC Curve Analysis for Recurrent Hip Fractures

The prediction probability model was formulated on the basis of multivariate regression, with the following formulation [24,25]: prediction probability P = e^x^/1 + e^x^, where e is the natural logarithm, X = −3.682 + 1.140 (age ≥ 85 years) + 1.333 (COPD) + 1.010 (PD) − 1.414 (antiosteoporosis drugs) − 0.682 (duration of antiosteoporosis drug therapy) − 0.884 (albumin ≥ 35 g·L^−1^). As demonstrated in Figure 2, ROC curve analysis was employed to evaluate the fitting effect between the predicted probability of recurrent hip fractures following primary osteoporotic hip fracture surgery. The area under the ROC curve (AUC) was 0.802 (95% CI = 0.743–0.862), with a *p* value of less than 0.0001. The optimal cut-off value was determined to be 0.533. At this juncture, the predictive sensitivity was found to be 77.8%, while the specificity was 75.5%. The findings suggested that the risk prediction model constructed according to the aforementioned factors exhibited adequate discriminant ability for the risk prediction of recurrent hip fractures following surgical treatment of primary osteoporotic hip fracture. The utilization of this risk prediction model has the potential to enhance patient recognition and medical provider decision-making, thereby facilitating more informed surgical observation and early screening for recurrent hip fractures following primary osteoporotic hip fracture surgery.

### 3.3. Comparison of Medical Costs and Outcomes for Patients with and Without Recurrent Hip Fractures

The 1-year, 2-year, and 3-year postoperative mortality rates were 5.9% (22/376), 8.5% (32/376), and 17.6% (66/376), respectively. The investigation revealed no statistically significant difference in total medical costs between patients with and without recurrent hip fractures (7835.9 ± 4261.9 vs. 7614.8 ± 3037.4 or median 6780.2 vs. 3383.6, *p* > 0.05). Moreover, no statistically significant difference in the duration of hospitalization was identified between the two groups (6.0 vs. 6.0, *p* > 0.05). Patients in the recurrent hip fractures group exhibited a higher two-year mortality rate (14.3% vs. 7.0%, *p* = 0.042) and a three-year mortality rate (26.0% vs. 15.4%, *p* = 0.029) (Table 3 and Figure 3). However, the study revealed no statistically significant difference in the three-year cumulative mortality risk between the recurrent hip fracture group and the non-recurrent hip fracture group (hazard ratio [HR] = 1.41, 95%CI = 0.83–2.38, *p* = 0.20) (Appendix A).

### 3.4. Risk Factors for Mortality in Older Patients Following Primary Osteoporotic Hip Fracture Surgery

In comparison with the living group, a statistically significant increase was observed in the number of patients aged ≥85 years in the death group (56.1% vs. 35.2%, *p* = 0.003). The utilization of antiosteoporosis drugs was observed to be more prevalent in the living group than in the death group (43.2% vs. 22.7%, *p* = 0.002). The median duration of antiosteoporosis drugs was found to be greater in the living group than in the death group (36.0 vs. 25.8, *p* < 0.001) (Appendix A). Multiple logistic regression analysis revealed that age ≥ 85 years (OR = 2.555, 95% CI = 1.473–4.433, *p* < 0.001) were identified as an independent risk factor for mortality. However, the use of antiosteoporosis drugs (OR = 0.369, 95%CI = 0.176–0.773, *p* = 0.008) and the duration of antiosteoporosis drug therapy (OR = 0.561, 95%CI = 0.328–0.895, *p* < 0.001) were identified as independent protective factors for mortality. The goodness of fit was subjected to evaluation using the Hosmer and Lemeshow analysis, from which a result of 0.758 was obtained (Table 4).

## 4. Discussion

Osteoporosis is a prevalent ailment among older adults. The diagnosis of an osteoporotic fracture is typically made when the trauma sustained is of a relatively low-energy nature, such as falling from a height or lower, and a force that would not be anticipated to result in such a fracture in a young, healthy adult [26]. Moreover, in light of the global increase in life expectancy, there is a mounting concern that the occurrence of hip fractures is set to rise exponentially with age, unless effective preventive measures are implemented [14]. However, it is noteworthy that the issue of recurrent hip fractures following surgical intervention for fragility fractures is of significant severity. Therefore, the identification of risk factors for recurrent hip fractures following surgical treatment of primary osteoporotic hip fractures in older patients will facilitate the establishment of a comprehensive, long-term and efficient prevention and treatment strategy for senile osteoporosis.

The present study found an incidence of recurrent hip fractures in older patients of 20.5% (77/376), which is comparatively lower than the incidence of refracture after a proximal femur fracture in older patients reported in another study (35.1%) [15]. As stated by Kaiwan Sriruanthong et al., a considerable proportion of second fractures (48%) occurred within a three-year period in patients aged 60 years or older who sustained osteoporotic fractures (hip, wrist, vertebra, and proximal humerus) from low-energy trauma [16]. The study demonstrated that patients within the recurrent hip fracture group exhibited a significantly higher mortality rate over a period of three years. The 3-year mortality rate following surgical treatment of primary osteoporotic hip fractures was found to be 17.6% (66/376). This figure is lower than the rate recorded in the previous study following intertrochanteric fracture surgery, which was 24.4% [27]. A meta-analysis of 75 studies involving a total of 64,316 patients revealed that the one-year and two-year mortality rates among hip fracture cases were 24.5% and 34.5%, respectively [28]. It is therefore strongly recommended that healthcare providers implement a comprehensive program with the objective of preventing subsequent fractures in patients who have sustained any form of primary fragility fracture.

Multiple logistic regression analysis revealed that age ≥ 85 years, COPD, and PD were identified as independent risk factors for recurrent hip fractures following surgical treatment of primary osteoporotic fractures in the present study. Conversely, the use of antiosteoporosis drugs, longer duration of antiosteoporosis drug therapy and serum albumin ≥ 35 g·L^−1^ were determined to be independent protective factors for recurrent hip fractures. As indicated by extant literature, the advanced age of subjects following a fragile fracture significantly increases the incidence of subsequent refractures [16,29,30,31]. In the report published by Kaiwan Sriruanthong et al., an increase adjusted odd ratio of 1.016 was reported for each year of age that exceeds 60 (1.17 for each decade) [16]. In the study conducted by J. Banefelt et al., it was reported that the odd ratio (OR) for patients aged 80 + years, in comparison with those aged between 50 and 59 years, was greater than 3.0 [31]. In comparison with patients in the 65–84 age group, a greater than 3.127-fold increase in OR values was recorded among patients in the 85+ age group in the present study. It has been established that older patients exhibit a diminished appetite. Consequently, patients afflicted with a hip fracture are more prone to malnutrition than the general population [32]. Some studies have indicated an association between albumin levels and mortality in patients suffering from hip fractures [27,33]. However, in contrast to the findings of the aforementioned studies, the investigation has revealed no association between albumin levels and mortality in patients who suffered from hip fractures. The findings of this study indicated that serum albumin levels of ≥35 g·L^−1^ functioned as an independent protective factor for recurrent hip fractures following primary osteoporotic hip fracture surgery. Consequently, orthopedic surgeons should meticulously monitor the perioperative nutrition management of older patients afflicted with osteoporotic hip fractures.

The present study identified COPD and PD to be independent risk factors for recurrent hip fractures following surgical treatment of primary osteoporotic hip fractures. Patients suffering from COPD exhibit a reduction in physical activity, resulting in a deterioration in their physical condition. This phenomenon consequently results in diminished bone quality and an elevated risk of falls [34]. COPD is a systemic disease that is complicated by various comorbidities, including OP. Furthermore, it is important to note that fractures associated with OP have the potential to result in a further deterioration of pulmonary function and impairment of activities of daily living. This, in turn, has the potential to engender a vicious cycle [35]. The study revealed that the recurrent hip fracture risk in older patients diagnosed with COPD following primary osteoporotic hip fracture surgery is approximately equivalent to 3.794 (95%CI = 1.747–8.236, *p* < 0.001). Moreover, an elevated prevalence of hip fracture has been documented in individuals diagnosed with PD. This phenomenon can be attributed to impaired gait, postural regulation, and cognitive dysfunction [28,36,37]. However, the risk of recurrent hip fractures among older adult patients afflicted with PD following surgical treatment of hip fractures remains to be thoroughly investigated. The recurrent hip fractures risk associated with PD, as determined in this study, was found to be almost equivalent to 2.744 (95%CI = 1.249–6.028, *p* = 0.012). Therefore, particular attention and care should be given to hip fracture patients who have comorbidities of COPD and PD. A primary focus should be placed on the prevention of falls, which may subsequently lead to fractures.

The present study identifies concomitant use of antiosteoporosis drugs and duration of antiosteoporosis drug therapy as independent protective factors. The findings provide a novel contribution to the extant literature on the subject, as previous reports have not explored this aspect. It has been asserted that there exists an “imminent risk” of further fractures within two years of an initial fragility fracture. It is imperative to acknowledge that this period signifies a pivotal stage in the treatment of osteoporosis [38]. As reported in the study by JenSheng Chen et al., the antiresorptive medication, Fosamax, has been demonstrated to enhance bone density and promote increased repair of bone tissue, thereby reducing the risk of further fractures [18]. As demonstrated in several research reports, denosumab has been demonstrated to decrease the risk of refracture and mortality in patients diagnosed with fragility fractures [17,20,21,39,40]. Despite the extensive utilization of bisphosphonates, denosumab, and teriparatide within the standard care paradigm for osteoporosis in China, the pharmacological treatment of osteoporotic fractures continues to be suboptimal [9]. It is imperative to acknowledge the pivotal role of medical intervention in the reduction in mortality and refracture risk. This underscores the necessity for further investigation and pharmacotherapy of osteoporosis, particularly within the context of fragility fractures.

The present study is subject to several limitations. First, the present study was conducted retrospectively and in a single-center study. Second, this study exclusively included older adult patients with osteoporotic hip fractures. Consequently, the findings may not be indicative of recurrent hip fractures in patients with traumatic or pathological fractures. Moreover, the study did not document the bone mineral density or biomarkers of the patients, primarily due to the absence of regular monitoring. Consequently, prospective studies will be designed to explore the inclusion of frailty indices or biomarkers, with a view to improving the accuracy of the models. Ultimately, the patients’ functional baseline, that is to say, their mobility and capacity for activities of daily living, remained unassessed. Notwithstanding, the study provides sufficient data to identify the risk factors for recurrent hip fractures following surgical treatment of primary osteoporotic hip fractures, thus providing a valuable foundation for the clinical practice of surgeons. The integration of this model into electronic medical records is envisaged as a future development, with the objective of facilitating automated risk scoring. The delivery of targeted interventions to high-risk older patients is to be accomplished through the conduction of dual-energy X-ray absorptiometry (DXA) screenings, the initiation of osteoporosis treatment, and the implementation of fall prevention programs.

## 5. Conclusions

Patients with advanced age, COPD and PD are more susceptible to recurrent hip fractures following surgical treatment of primary osteoporotic hip fractures and should be prioritized for increased care and monitoring. Furthermore, the management of potentially modifiable risk factors constitutes a significant adjunct to the pharmacologic osteoporosis therapy to forestall the occurrence of recurrent hip fractures following primary osteoporotic hip fracture surgery, thereby reducing mortality rate. Therefore, the present study may also provide a theoretical foundation for clinical diagnostics, the integration of public health policy, and the validation of biomarkers.

## Figures and Tables

**Figure 1 diseases-13-00351-f001:**
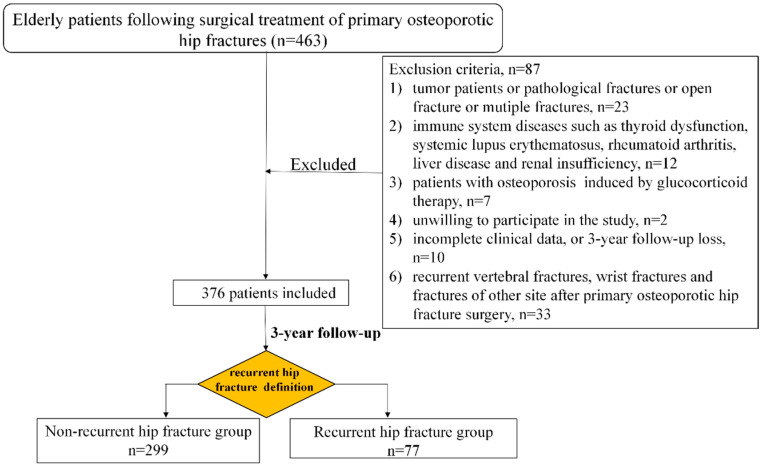
Flowchart of the study.

**Figure 2 diseases-13-00351-f002:**
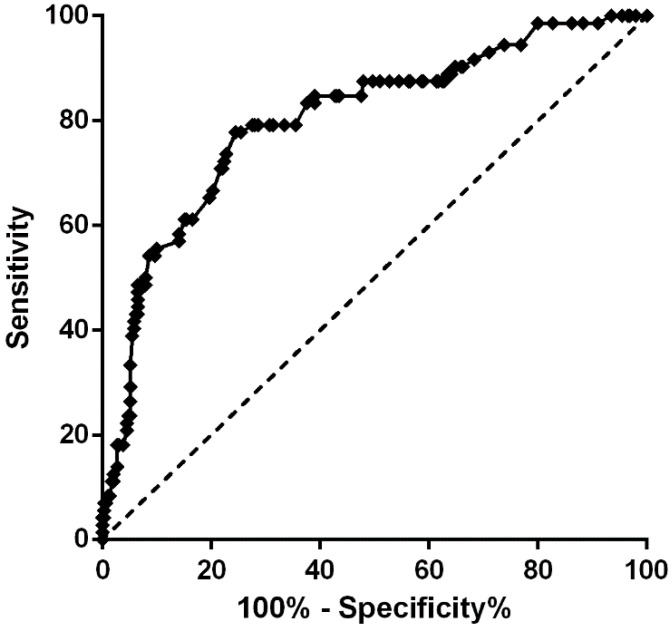
Receiver operating characteristic (ROC) curve for the model of risk factors; *p* < 0.001, AUC = 0.802 (95% CI 0.743~0.862). Logit (P) = −3.682 + 1.140 [Age ≥ 85 years] + 1.333 [COPD] + 1.010 [PD] − 1.414 [Antiosteoporosis drugs] − 0.682 [Duration of antiosteoporosis drug therapy] − 0.884 [Serum albumin ≥ 35 g·L^−1^]. Moreover, the predictive sensitivity was 77.8%, and the specificity was 75.5%.

**Figure 3 diseases-13-00351-f003:**
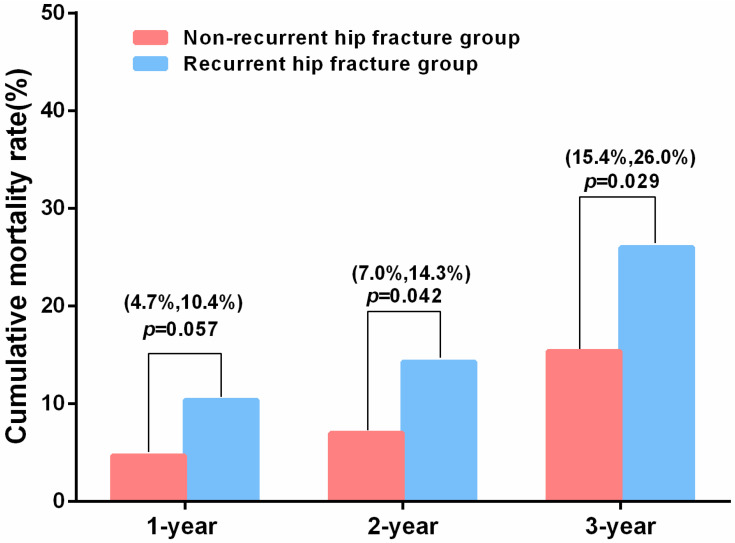
Cumulative mortality rate between the non-recurrent hip fracture group and the recurrent hip fracture group.

**Table 1 diseases-13-00351-t001:** Demographic information and clinical characteristics of patients following surgical treatment of primary osteoporotic hip fractures between non-recurrent hip fracture group and recurrent hip fracture group.

	Total (n = 376)	Non-Recurrent Hip Fracture Group (n = 299)	Recurrent Hip Fracture Group (n = 77)	*p* Value
Gender, n (%)				0.247
female	289 (76.9)	226 (75.6)	63 (81.8)	
male	87 (23.1)	73 (24.4)	14 (18.2)	
Age, n (%)				0.002
65–84 years	242 (64.4)	204 (68.2)	38 (49.4)	
≥85 years	134 (35.6)	95 (31.8)	39 (50.6)	
BMI (kg·m^−2^), n (%)				0.474
<19	64 (17.0)	53 (17.7)	11 (14.3)	
≥19	312 (83.0)	246 (82.3)	66 (85.7)	
ASA, n (%)				0.098
1	112 (29.8)	85 (28.4)	27 (35.1)	
2	240 (63.8)	198 (66.2)	42 (54.5)	
3	24 (6.4)	16 (5.4)	8 (10.4)	
Site of primary hip fracture, n (%)				0.110
Left	185 (49.2)	155 (51.8)	30 (39.0)	
Right	191 (50.8)	144 (48.2)	47 (61.0)	
Surgical types of primary hip fracture, n (%)				0.969
Internal fixation	157 (41.8)	125 (41.8)	32 (41.6)	
Hip arthroplasty	219 (58.2)	174 (58.2)	45 (58.4)	
Types of primary hip fracture, n (%)				0.774
Femoral neck fracture	259 (68.9)	207 (69.2)	52 (67.5)	
intertrochanteric fracture	117 (31.1)	92 (30.8)	25 (32.5)	
Anesthesia, n (%)				0.324
General anesthesia	73 (19.4)	55 (18.4)	59 (76.6)	
Spinal anesthesia	303 (80.6)	244 (81.6)	18 (23.4)	
Smoke, n (%)	37 (9.8)	29 (9.7)	8 (10.4)	0.682
Alcohol, n (%)	57 (15.2)	37 (12.4)	20 (26.0)	0.015
Blood transfusion	81 (21.5)	62 (20.7)	19 (24.7)	0.453
Concomitant underlying diseases				
Hypertension, n (%)	163 (43.3)	132 (44.1)	31 (40.3)	0.539
Diabetes, n (%)	71 (18.9)	62 (20.7)	9 (11.7)	0.070
Heart attack, n (%)	64 (17.0)	14 (18.2)	50 (16.7)	0.761
Stroke, n (%)	35 (9.3)	30 (10.0)	5 (6.5)	0.340
Heart failure, n (%)	7 (1.9)	4 (1.3)	3 (3.9)	0.139
COPD, n (%)	48 (12.8)	28 (9.4)	20 (26.0)	<0.001
Anxiety/Depression, n (%)	5 (1.3)	4 (1.3)	1 (1.3)	0.979
Dementia, n (%)	17 (4.5)	9 (3.0)	8 (10.4)	0.005
PD, n (%)	45 (12.0)	26 (8.7)	19 (24.7)	<0.001
Deep vein thrombosis, n (%)	11 (2.9)	9 (3.0)	2 (2.6)	0.848
Sleep disturbance, n (%)	8 (2.1)	6 (2.0)	2 (2.6)	0.749
Antiosteoporosis drugs, n (%)	149 (39.6)	134 (44.8)	15 (19.5)	<0.001
Duration of antiosteoporosis drug therapy (month) Median (IQR)	34.5 (9.4)	36.0 (3.1)	33.9 (10.1)	<0.001
Laboratory variables				
Hb level at admission (g·L^−1^) Mean ± SD	105.7 ± 17.5	106.0 ± 17.3	104.4 ± 18.4	0.490
Albumin (g·L^−1^), n (%)				0.010
≥35	108 (29.3)	95 (32.4)	13 (17.3)	
<35	260 (70.7)	198 (67.6)	62 (82.7)	
eGFR(mL·min^−1^·1.73^−2^) Median (IQR)	76.0 (25.8)	75.5 (27)	83.0 (22.0)	0.486
Serum calcium (mmol·L^−1^) Mean ± SD	2.17 ± 0.14	2.17 ± 0.13	2.11 ± 0.16	0.057
25-hydroxyvitamin D (nmol·L^−1^) Median (IQR)	32.1 (22.1)	33.1 (22.7)	26.0 (23.3)	0.173

Abbreviations: BMI, body mass index; ASA, American society of anesthesiologists; COPD, chronic obstructive pulmonary disease; PD, Parkinson’s disease; Hb, hemoglobin; eGFR, estimated Glomerular Filtration Rate; IQR, interquartile range; SD, standard deviation.

**Table 2 diseases-13-00351-t002:** Multivariable logistic regression of factors for recurrent hip fractures following surgical treatment of primary osteoporotic hip fractures.

	B	OR	95%CI		*p* Value
			Lower	Upper	
≥85 years	1.140	3.127	1.672	5.849	<0.001
COPD	1.333	3.794	1.747	8.236	<0.001
PD	1.010	2.744	1.249	6.028	0.012
Antiosteoporosis drugs	−1.414	0.243	0.131	0.451	<0.001
Duration of antiosteoporosis drugs therapy	−0.682	0.564	0.283	0.830	0.003
Albumin ≥ 35 g·L^−1^	−0.884	0.413	0.194	0.881	0.022
Constant	−3.682	0.025			<0.001

Abbreviations: COPD, chronic obstructive pulmonary disease; PD, Parkinson’s disease; B, beta; OR, odd ratios; CI, confidence interval.

**Table 3 diseases-13-00351-t003:** Medical costs and outcomes of patients with and without recurrent hip fractures following surgical treatment of primary osteoporotic hip fractures.

	Total (n = 376)	Non-Recurrent Hip Fracture Group (n = 299)	Recurrent Hip Fracture Group (n = 77)	*p* Value
Total costs (US $) Median (IQR)	7085.6 (3590.6)	7149.4 (3383.6)	6780.2 (2876.1)	0.782
Length of hospital stay(day) Median (IQR)	6.0 (5.0)	6.0 (5.0)	6.0 (6.0)	0.661
Cumulative 1-year mortality, n (%)	22 (5.9)	14 (4.7)	8 (10.4)	0.057
Cumulative 2-year mortality, n (%)	32 (8.5)	21 (7.0)	11 (14.3)	0.042
Cumulative 3-year mortality, n (%)	66 (17.6)	46 (15.4)	20 (26.0)	0.029

**Table 4 diseases-13-00351-t004:** Multivariable logistic regression of factors for mortality following surgical treatment of primary osteoporotic hip fractures.

	B	OR	95%CI		*p* Value
			Lower	Upper	
≥85 years	0.938	2.555	1.473	4.433	<0.001
Antiosteoporosis drugs	−1.036	0.369	0.176	0.773	0.008
Duration of antiosteoporosis drug therapy	−0.150	0.561	0.328	0.895	<0.001
Constant	2.003	7.414			<0.001

Abbreviations: B, beta; OR, odd ratios; CI, confidence interval.

## Data Availability

Access to the relevant datasets and the analytical methods employed in the present study are available upon reasonable request from the corresponding author.

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
