# Peer review of "Risk Factors for Recurrent Hip Fractures Following Surgical Treatment of Primary Osteoporotic Hip Fractures in Chinese Older Adults"

_diseases, 2025, doi:10.3390/diseases13110351_

Round 1
Reviewer 1 Report
Comments and Suggestions for Authors
The manuscript is clearly written and logically structured, with strong potential for clinical application in risk stratification and secondary prevention strategies. However, several major and minor issues must be addressed:
Avoid the use of elderly in scientific paper. In place, use “older adults”
Abtract
According journal guideline “Systematic reviews and original research articles should have a structured abstract of around 250 words and contain the following headings: Background/Objectives, Methods, Results, and Conclusions”. Please included all the headings
Specify the study design: “single-center, retrospective cohort study.”
Clarify the follow-up duration
State how recurrent hip fracture was defined and confirmed
In results: Include 95% confidence intervals for all odds ratios.
Avoid overstatement. Instead of “can effectively predict,”
Introduction
Explicitly state that while many studies have examined first hip fractures, fewer have focused on recurrence risk, especially in Asian populations.
Emphasize that existing prediction models may not adequately incorporate neurological comorbidities like PD and dementia. For example, Q-fracture included
Briefly explain why PD/dementia is a plausible un-explore risk factor
Methods
Clarify whether patients were followed prospectively after their first fracture or if the entire cohort was identified retrospectively.
Clarify how comorbidities were diagnosed: chart review, ICD codes, physician diagnosis?
Specify the duration variable: is it time from first fracture to start of anti-osteoporosis therapy?
Results
Define abbreviations at first use
In ROC analysis, interpret the optimal cutoff point (Youden index) and its sensitivity/specificity.
Discussion
Emphasize that lack of osteoporosis treatment is the strongest protective factor absent
Explain why COPD and dementia are risk factors:
Reference studies showing under-treatment of osteoporosis post-fracture (the “treatment gap”).
Cite meta-analyses on neurological diseases and fracture risk.
Suggest integrating the model into electronic health records for automated risk scoring.
Propose targeted interventions: DXA screening, osteoporosis therapy initiation, fall prevention programs for high-risk individuals.
Explore adding frailty indices or biomarkers to improve model accuracy.
Author Response
1. The manuscript is clearly written and logically structured, with strong potential for clinical application in risk stratification and secondary prevention strategies. However, several major and minor issues must be addressed:
Avoid the use of elderly in scientific paper. In place, use “older adults”
Response: We are extremely grateful for the constructive comments provided on this manuscript.
Line 2: We have revised the title of the manuscript to “Risk Factors for Recurrent Hip Fractures Following Surgical Treatment of Primary Osteoporotic Hip Fractures in Chinese Older Adults”.
2. Abstract
2.1 According journal guideline “Systematic reviews and original research articles should have a structured abstract of around 250 words and contain the following headings: Background/Objectives, Methods, Results, and Conclusions”. Please included all the headings
Specify the study design: “single-center, retrospective cohort study.”
Response: We appreciate your valuable comments. We have made revisions in the methods section of the Abstract, which are marked in red font.
Lines 32-34: A single-center, retrospective cohort study was conducted on 376 patients suffering from primary osteoporotic hip fractures from January 1, 2020 to December 31, 2021.
2.2 Clarify the follow-up duration
Response: We are indebted to you for your question. We have made revisions and provided detailed explanations in the second paragraph of the materials and methods section, which are marked in red font.
Lines 35-36: In this study, 376 patients were observed over a three-year period.
Lines 121-133: The present study utilized a three-year period of follow-up data to assess the incidence of recurrent hip fractures subsequent to primary osteoporotic fractures. A comprehensive review of the patients' prescriptions and medications was conducted during the enrolment and follow-up procedures, with meticulous documentation being carried out accordingly. Follow-up time was defined as the number of days from the initial fracture event to the occurrence of recurrent hip fracture or the most recent appointment, whichever occurred first. The term “recurrent hip fracture” was defined as the time to the first occurrence after primary osteoporotic hip fracture surgery, with examples including ipsilateral or contralateral hip fractures. The diagnosis of hip fracture is made on the basis of the results of diagnostic imaging. The primary endpoint and statistical analysis were predetermined in advance. The primary endpoint of the study was defined as the timeframe to the initial occurrence of a subsequent hip fracture. The occurrence of mortality following surgical treatment of primary osteoporotic hip fractures was identified as the secondary endpoint.
2.3 State how recurrent hip fracture was defined and confirmed
Response: I would like to express my gratitude for your question. We have made revisions in the Methods section of the abstract, which are marked in red font.
Lines 126-129:The term “recurrent hip fracture” was defined as the time to the first occurrence after primary osteoporotic hip fracture surgery, with examples including ipsilateral or contralateral hip fractures. The diagnosis of hip fracture is made on the basis of the results of diagnostic imaging.
2.4 In results: Include 95% confidence intervals for all odds ratios.
Avoid overstatement. Instead of “can effectively predict,”
Response: We appreciate your valuable comments. We have made revisions in the methods section of the Abstract, which are marked in red font.
Lines 37-43:We have revised this sentence. Multiple logistic regression analysis revealed that age≥85 years (odd ratios [OR]=3.127, 95% confidence interval [CI]=1.672-5.849, p<0.001), chronic obstructive pulmonary disease (COPD) (OR=3.794, 95%CI=1.747-8.236, p<0.001), and Parkinson's disease (PD) (OR=2.744, 95%CI=1.249-6.028, p=0.012) were independent risk factors for recurrent hip fractures; antiosteoporosis drugs (OR=0.243, 95%CI=0.131-0.451, p<0.001), duration of antiosteoporosis drug therapy (OR = 0.564, 95%CI=0.283-0.830, p=0.003) and serum albumin≥35 g·L-1 (OR=0.413, 95%CI=0.194-0.881, p=0.022) were independent protective factors for recurrent hip fractures.
3. Introduction
3.1 Explicitly state that while many studies have examined first hip fractures, fewer have focused on recurrence risk, especially in Asian populations.
Response: We appreciate your valuable comments. We have made revisions in the second paragraph of the Introduction section and marked them in red font.
Lines 76-79: Despite the fact that prior research has investigated the risk factors for refracture among older patients following hip fractures [15, 16], studies on the risk of recurrent hip fractures following surgical treatment of primary osteoporotic hip fractures are relatively scarce, particularly in the Asian population.
3.2 Emphasize that existing prediction models may not adequately incorporate neurological comorbidities like PD and dementia. For example, Q-fracture included
Briefly explain why PD/dementia is a plausible un-explore risk factor
Response: We are grateful for your constructive feedback. We have made revisions in the third paragraph of the Introduction section and marked them in red font.
Lines 90-94: Moreover, existing prediction models may not fully incorporate neurological comorbidities among Chinese older patients, such as Parkinson's disease and dementia. Patients suffering from Parkinson's disease and dementia are predisposed to a heightened risk of falling, a phenomenon that can be attributed to impaired gait, postural regulation, and cognitive impairment [23]. Consequently, these factors may contribute to an elevated risk of fracture.
4. Methods
4.1 Clarify whether patients were followed prospectively after their first fracture or if the entire cohort was identified retrospectively.
Response: We appreciate your valuable comments. We have made revisions in the first paragraph of the Materials and Methods section and marked them in red font.
Lines 114-116: In total, 376 out of 463 patients met the inclusion criteria and were subjected to retrospective follow-up over a period of three years following surgical treatment of primary osteoporotic hip fractures.
4.2 Clarify how comorbidities were diagnosed: chart review, ICD codes, physician diagnosis?
Response: I would like to take this opportunity to express my gratitude for your enquiry. We have made revisions in the first paragraph of the Materials and Methods section and marked them in red font.
Lines 103-105: The diagnosis of comorbidities in patients is based on the 10th Edition of the International Classification of Diseases (ICD-10).
4.3 Specify the duration variable: is it time from first fracture to start of anti-osteoporosis therapy?
Response: I would like to express my gratitude for your question. We have made revisions and provided detailed explanations in the second paragraph of the Materials and Methods section, which are marked in red font.
Lines 120-124: The present study population was comprised of patients diagnosed with a hip fracture that met the criteria for osteoporosis. The present study utilized a three-year period of follow-up data to assess the incidence of recurrent hip fractures subsequent to primary osteoporotic fractures. A comprehensive review of the patients' prescriptions and medications was conducted during the enrolment and follow-up procedures, with meticulous documentation being carried out accordingly.
5. Results
5.1 Define abbreviations at first use
Response: We appreciate your valuable comments. We have made revisions to the first use of abbreviations in the manuscript.
5.2 In ROC analysis, interpret the optimal cutoff point (Youden index) and its sensitivity/specificity.
Response: We are grateful for your constructive feedback. We have provided explanations and made revisions in the ROC curve analysis for recurrent hip fractures section of the Results section, which are marked in red font.
Lines 234-240: The findings suggested that the risk prediction model constructed according to the aforementioned factors exhibited adequate discriminant ability for the risk prediction of recurrent hip fractures following surgical treatment of primary osteoporotic hip fracture. The utilization of this risk prediction model has the potential to enhance patient recognition and medical provider decision-making, thereby facilitating more informed surgical observation and early screening for recurrent hip fractures following primary osteoporotic hip fracture surgery.
6. Discussion
6.1 Emphasize that lack of osteoporosis treatment is the strongest protective factor absent
Response: We are grateful for your constructive feedback.
Lines 345-359:we have rewritten this section in the fifth paragraph of the Discussion section.
The present study identifies concomitant use of antiosteoporosis drugs and duration of antiosteoporosis drug therapy as independent protective factors. The findings provide a novel contribution to the extant literature on the subject, as previous reports have not explored this aspect. It has been asserted that there exists an "imminent risk" of further fractures within two years of an initial fragility fracture. It is imperative to acknowledge that this period signifies a pivotal stage in the treatment of osteoporosis [38]. As reported in the study by JenSheng Chen et al., the antiresorptive medication, Fosamax, has been demonstrated to enhance bone density and promote increased repair of bone tissue, thereby reducing the risk of further fractures [18]. As demonstrated in several research reports, denosumab has been demonstrated to decrease the risk of refracture and mortality in patients diagnosed with fragility fractures [17, 20, 21, 39, 40]. Despite the extensive utilization of bisphosphonates, denosumab, and teriparatide within the standard care paradigm for osteoporosis in China, the pharmacological treatment of osteoporotic fractures continues to be suboptimal [9]. It is imperative to acknowledge the pivotal role of medical intervention in the reduction of mortality and refracture risk. This underscores the necessity for further investigation and pharmacotherapy of osteoporosis, particularly within the context of fragility fractures.
6.2 Explain why COPD and dementia are risk factors:
Response: We appreciate your valuable comments.
Lines 327-344:We have discussed COPD and PD as risk factors in the fourth paragraph of the Discussion section. we have rewritten this section. The present study identified COPD and PD to be independent risk factors for recurrent hip fractures following surgical treatment of primary osteoporotic hip fractures. Patients suffering from COPD exhibit a reduction in physical activity, resulting in a deterioration in their physical condition. This phenomenon consequently results in diminished bone quality and an elevated risk of falls [34]. COPD is a systemic disease that is complicated by various comorbidities, including OP. Furthermore, it is important to note that fractures associated with OP have the potential to result in a further deterioration of pulmonary function and impairment of activities of daily living. This, in turn, has the potential to engender a vicious cycle [35]. The study revealed that the recurrent hip fracture risk in older patients diagnosed with COPD following primary osteoporotic hip fracture surgery is approximately equivalent to 3.794 (95%CI=1.747-8.236, p<0.001). Moreover, an elevated prevalence of hip fracture has been documented in individuals diagnosed with PD. This phenomenon can be attributed to impaired gait, postural regulation, and cognitive dysfunction [28, 36, 37]. However, the risk of recurrent hip fractures among older adult patients afflicted with PD following surgical treatment of hip fractures remains to be thoroughly investigated. The recurrent hip fractures risk associated with PD, as determined in this study, was found to be almost equivalent to 2.744 (95%CI=1.249-6.028, p=0.012). Therefore, particular attention and care should be given to hip fracture patients who have comorbidities of COPD and PD. A primary focus should be placed on the prevention of falls, which may subsequently lead to fractures.
6.3 Reference studies showing under-treatment of osteoporosis post-fracture (the “treatment gap”).
Response: We appreciate your valuable comments. We have made revisions in the fifth paragraph of the Discussion, and marked them in red font.
Lines 354-357: Despite the extensive utilization of bisphosphonates, denosumab, and teriparatide within the standard care paradigm for osteoporosis in China, the pharmacological treatment of osteoporotic fractures continues to be suboptimal.
6.4 Cite meta-analyses on neurological diseases and fracture risk.
Response: We highly appreciate the valuable suggestions you have provided. We have added references in the fourth paragraph of the Discussion section, and marked them in red font.
Lines 336-338: Moreover, an elevated prevalence of hip fracture has been documented in individuals diagnosed with PD. This phenomenon can be attributed to impaired gait, postural regulation, and cognitive dysfunction [28, 36, 37].
6.5 Suggest integrating the model into electronic health records for automated risk scoring.
Propose targeted interventions: DXA screening, osteoporosis therapy initiation, fall prevention programs for high-risk individuals.
Explore adding frailty indices or biomarkers to improve model accuracy.
Response: We are extremely grateful for the valuable suggestions you have provided. We have made revisions in the sixth paragraph of the Discussion section and marked them in red font.
Lines 363-366: Moreover, the study did not document the bone mineral density or biomarkers of the patients, primarily due to the absence of regular monitoring. Consequently, prospective studies will be designed to explore the inclusion of frailty indices or biomarkers, with a view to improving the accuracy of the models.
Lines 370-375: The integration of this model into electronic medical records is envisaged as a future development, with the objective of facilitating automated risk scoring. The delivery of targeted interventions to high-risk older patients is to be accomplished through the conduction of dual-energy X-ray absorptiometry (DXA) screenings, the initiation of osteoporosis treatment, and the implementation of fall prevention programs.

Reviewer 2 Report
Comments and Suggestions for Authors
- Table 1 “Side” contains inconsistencies. Please recheck the raw data, correct the table, and report revised p-values if they change.
- The outcome (recurrent hip fracture) is analyzed as a 3-year yes/no variable using logistic regression. In an elderly cohort, death is a clear competing event for subsequent fracture; ignoring it likely biases effect estimates. I therefore recommend the authors (a) report individual follow-up times and numbers censored/dead, (b) use time-to-event analysis with competing-risk methods (cumulative incidence function + Fine–Gray model) as the primary analysis, and (c) provide cause-specific Cox results as a supplementary analysis. Please also add CIF plots and numbers-at-risk (events/deaths/censored) at 1, 2, and 3 years.
- Line 184 / ref.16: although the cited paper contains the formula, that clinical study is not an appropriate methodological reference for logistic regression. Please replace or supplement ref.16 with a standard statistical reference (e.g. Hosmer & Lemeshow or Steyerberg/TRIPOD).
- The albumin variable has been dichotomized at 35 g/L. Please justify this cutoff.
- Please report timing of recurrent fractures. This information is essential to interpret the clinical implications.
- The definition of “recurrent hip fracture” is unclear. State explicitly whether events include contralateral fractures only, ipsilateral implant-related/periprosthetic fractures, or both. If implant-related fractures are included, please provide a breakdown (ipsilateral periprosthetic vs ipsilateral non-periprosthetic vs contralateral).
- Line 196 contains a typo: ship fracture.
- Methods lack detail on cost measurement despite reporting cost outcomes. Please specify perspective, data source, items included, currency and price year, and whether costs are per patient cumulative over follow-up or per admission. Given skewness of cost data, report both mean (SD) and median (IQR) and clarify statistical methods used for cost comparisons.
Author Response
Comments and Suggestions for Authors
- Table 1 “Side” contains inconsistencies. Please recheck the raw data, correct the table, and report revised p-values if they change.
Response: We appreciate your valuable comments. We have carefully reviewed and revised Table 1.
Table 1 Demographic information and clinical characteristics of patients following surgical treatment of primary osteoporotic hip fractures between non-recurrent hip fracture group and recurrent hip fracture group.
|
|
Total (n=376) |
Non-recurrent hip fracture group (n=299) |
Recurrent hip fracture group (n=77) |
p value |
|
Gender, n (%) |
|
|
|
0.247 |
|
female |
289(76.9) |
226(75.6) |
63(81.8) |
|
|
male |
87(23.1) |
73(24.4) |
14(18.2) |
|
|
Age, n (%) |
|
|
|
0.002 |
|
65-84 years |
242(64.4) |
204(68.2) |
38(49.4) |
|
|
≥85 years |
134(35.6) |
95(31.8) |
39(50.6) |
|
|
BMI (kg·m-2), n (%) |
|
|
|
0.474 |
|
<19 |
64(17.0) |
53(17.7) |
11(14.3) |
|
|
≥19 |
312(83.0) |
246(82.3) |
66(85.7) |
|
|
ASA, n (%) |
|
|
|
0.098 |
|
1 |
112(29.8) |
85(28.4) |
27(35.1) |
|
|
2 |
240(63.8) |
198(66.2) |
42(54.5) |
|
|
3 |
24(6.4) |
16(5.4) |
8(10.4) |
|
|
Site of primary hip fracture, n (%) |
|
|
|
0.110 |
|
Left |
185(49.2) |
155(51.8) |
30(39.0) |
|
|
Right |
191(50.8) |
144(48.2) |
47(61.0) |
|
|
Surgical types of primary hip fracture, n (%) |
|
|
|
0.969 |
|
Internal fixation |
157(41.8) |
125(41.8) |
32(41.6) |
|
|
Hip arthroplasty |
219(58.2) |
174(58.2) |
45(58.4) |
|
|
Types of primary hip fracture, n (%) |
|
|
|
0.774 |
|
Femoral neck fracture |
259(68.9) |
207(69.2) |
52(67.5) |
|
|
intertrochanteric fracture |
117(31.1) |
92(30.8) |
25(32.5) |
|
|
Anesthesia, n (%) |
|
|
|
0.324 |
|
General anesthesia |
73(19.4) |
55(18.4) |
59(76.6) |
|
|
Spinal anesthesia |
303(80.6) |
244(81.6) |
18(23.4) |
|
|
Smoke, n (%) |
37(9.8) |
29(9.7) |
8(10.4) |
0.682 |
|
Alcohol, n (%) |
57(15.2) |
37(12.4) |
20(26.0) |
0.015 |
|
Blood transfusion |
81(21.5) |
62(20.7) |
19(24.7) |
0.453 |
|
Concomitant underlying diseases |
|
|
|
|
|
Hypertension, n (%) |
163(43.3) |
132(44.1) |
31(40.3) |
0.539 |
|
Diabetes, n (%) |
71(18.9) |
62(20.7) |
9(11.7) |
0.070 |
|
Heart attack, n (%) |
64(17.0) |
14(18.2) |
50(16.7) |
0.761 |
|
Stroke, n (%) |
35(9.3) |
30(10.0) |
5(6.5) |
0.340 |
|
Heart failure, n (%) |
7(1.9) |
4(1.3) |
3(3.9) |
0.139 |
|
COPD, n (%) |
48(12.8) |
28(9.4) |
20(26.0) |
<0.001 |
|
Anxiety/Depression, n (%) |
5(1.3) |
4(1.3) |
1(1.3) |
0.979 |
|
Dementia, n (%) |
17(4.5) |
9(3.0) |
8(10.4) |
0.005 |
|
PD, n (%) |
45(12.0) |
26(8.7) |
19(24.7) |
<0.001 |
|
Deep vein thrombosis, n (%) |
11(2.9) |
9(3.0) |
2(2.6) |
0.848 |
|
Sleep disturbance, n (%) |
8(2.1) |
6(2.0) |
2(2.6) |
0.749 |
|
Antiosteoporosis drugs, n (%) |
149(39.6) |
134(44.8) |
15(19.5) |
<0.001 |
|
Duration of antiosteoporosis drug therapy (month) Median (IQR) |
34.5(9.4) |
36.0(3.1) |
33.9(10.1) |
<0.001 |
|
Laboratory variables |
|
|
|
|
|
Hb level at admission (g·L-1) Mean±SD |
105.7±17.5 |
106.0±17.3 |
104.4±18.4 |
0.490 |
|
Albumin (g·L-1) , n (%) |
|
|
|
0.010 |
|
≥35 |
108(29.3) |
95(32.4) |
13(17.3) |
|
|
<35 |
260(70.7) |
198(67.6) |
62(82.7) |
|
|
eGFR(ml·min-1·1.73-2) Median (IQR) |
76.0(25.8) |
75.5(27) |
83.0(22.0) |
0.486 |
|
Serum calcium (mmol·L-1) Mean±SD |
2.17±0.14 |
2.17±0.13 |
2.11±0.16 |
0.057 |
|
25-hydroxyvitamin D (nmol·L-1) Median (IQR) |
32.1(22.1) |
33.1(22.7) |
26.0(23.3) |
0.173 |
Abbreviations: BMI, body mass index; ASA, American society of anesthesiologists; COPD, chronic obstructive pulmonary disease; PD, Parkinson’s disease; Hb, hemoglobin; eGFR, estimated Glomerular Filtration Rate; IQR, interquartile range; SD, standard deviation.
- The outcome (recurrent hip fracture) is analyzed as a 3-year yes/no variable using logistic regression. In an elderly cohort, death is a clear competing event for subsequent fracture; ignoring it likely biases effect estimates. I therefore recommend the authors (a) report individual follow-up times and numbers censored/dead, (b) use time-to-event analysis with competing-risk methods (cumulative incidence function + Fine–Gray model) as the primary analysis, and (c) provide cause-specific Cox results as a supplementary analysis. Please also add CIF plots and numbers-at-risk (events/deaths/censored) at 1, 2, and 3 years.
Response: We are extremely grateful for the valuable suggestions you have provided. We have made revisions in the patient outcomes section (Comparison of medical costs and outcomes for patients with and without recurrent hip fractures) of the Results section and marked them in red font.
Lines 256-258: However, the study revealed no statistically significant difference in the three-year cumulative mortality risk between the recurrent hip fracture group and the non-recurrent hip fracture group (hazard ratio [HR]=1.41, 95%CI=0.83-2.38, p=0.20) (Supplementary Figure S1).
- Line 184 / ref.16: although the cited paper contains the formula, that clinical study is not an appropriate methodological reference for logistic regression. Please replace or supplement ref.16 with a standard statistical reference (e.g. Hosmer & Lemeshow or Steyerberg/TRIPOD).
Response: We appreciate your valuable comments. We have made revisions to this section.
Lines 226-227: The prediction probability model was formulated on the basis of multivariate regression, with the following formulation [24, 25]:
- The albumin variable has been dichotomized at 35 g/L. Please justify this cutoff.
Response: We are extremely grateful for the valuable suggestions you have provided. Clinically, a serum albumin level below 35 g/L is diagnosed as hypoalbuminemia; therefore, 35 g/L was chosen as the cut-off value.
- Please report timing of recurrent fractures. This information is essential to interpret the clinical implications.
Response: We appreciate your valuable comments. We have made revisions in the first paragraph of the Results and marked them in red font.
Lines 183-184: The median time to recurrent hip fractures was 22.4 months, with an interquartile range (IQR) of 23.8 months.
- The definition of “recurrent hip fracture” is unclear. State explicitly whether events include contralateral fractures only, ipsilateral implant-related/periprosthetic fractures, or both. If implant-related fractures are included, please provide a breakdown (ipsilateral periprosthetic vs ipsilateral non-periprosthetic vs contralateral).
Response: We are extremely grateful for the valuable suggestions you have provided. We have made revisions in the first paragraph of the Materials and Methods section and marked them in red font.
Lines 126-129: The term “recurrent hip fracture” was defined as the time to the first occurrence after primary osteoporotic hip fracture surgery, with examples including ipsilateral or contralateral hip fractures. Because the number of periprosthetic fracture cases was small, this study did not include them.
- Line 196 contains a typo: ship fracture.
Response: We appreciate your valuable comments. We have made revisions to this section.
- Methods lack detail on cost measurement despite reporting cost outcomes. Please specify perspective, data source, items included, currency and price year, and whether costs are per patient cumulative over follow-up or per admission. Given skewness of cost data, report both mean (SD) and median (IQR) and clarify statistical methods used for cost comparisons.
Response: We are extremely grateful for the valuable suggestions you have provided. We have made revisions in the patient outcomes section (Comparison of medical costs and outcomes for patients with and without recurrent hip fractures) of the Results section and marked them in red font.
Lines 250-254: The investigation revealed no statistically significant discrepancy in total medical costs between patients with and without recurrent hip fractures (7835.9±4261.9 vs. 7614.8±3037.4 or median 6780.2 vs. 3383.6, p>0.05). Moreover, no statistically significant difference in the duration of hospitalization was identified between the two groups (6.0 vs. 6.0, p>0.05).

Reviewer 3 Report
Comments and Suggestions for Authors
The manuscript addresses an important biomedical problem; however, the novelty compared to previously published studies should be more clearly articulated in the Introduction.
1) The motivation and clinical significance are strong, but could be supported by recent epidemiological statistics to emphasise the global health relevance.
2) Certain experimental procedures lack sufficient detail
3) Most of the figures have low resolution or small fonts. Please check
4) Include a critical comparison with at least three recent papers
5) Add a concise paragraph explicitly discussing the study’s limitations
6) The paper is generally well-written but would benefit from minor grammatical polishing and simplification of long sentences.
7) Expand the conclusion to outline potential translational applications, such as clinical diagnostics, public health policy integration, or biomarker validation studies.
Author Response
Comments and Suggestions for Authors
The manuscript addresses an important biomedical problem; however, the novelty compared to previously published studies should be more clearly articulated in the Introduction.
1. The motivation and clinical significance are strong, but could be supported by recent epidemiological statistics to emphasise the global health relevance.
Response: We are extremely grateful for the valuable suggestions you have provided. We have made revisions in the first paragraph of the Introduction section and marked them in red font.
Lines 54-57: As the global population continues to age, there is an increasing prevalence of osteoporosis and the subsequent occurrence of osteoporotic fractures. This situation gives rise to a marked menace with regard to public health [2-5].
2. Certain experimental procedures lack sufficient detail
Response: We appreciate your valuable comments. We have made revisions in the second paragraph of the Materials and Methods section and marked them in red font.
Lines 120-133: The present study population was comprised of patients diagnosed with a hip fracture that met the criteria for osteoporosis. The present study utilized a three-year period of follow-up data to assess the incidence of recurrent hip fractures subsequent to primary osteoporotic fractures. A comprehensive review of the patients' prescriptions and medications was conducted during the enrolment and follow-up procedures, with meticulous documentation being carried out accordingly. Follow-up time was defined as the number of days from the initial fracture event to the occurrence of recurrent hip fracture or the most recent appointment, whichever occurred first. The term “recurrent hip fracture” was defined as the time to the first occurrence after primary osteoporotic hip fracture surgery, with examples including ipsilateral or contralateral hip fractures. The diagnosis of hip fracture is made on the basis of the results of diagnostic imaging. The primary endpoint and statistical analysis were predetermined in advance. The primary endpoint of the study was defined as the timeframe to the initial occurrence of a subsequent hip fracture. The occurrence of mortality following surgical treatment of primary osteoporotic hip fractures was identified as the secondary endpoint.
3. Most of the figures have low resolution or small fonts. Please check
Response: We appreciate your valuable comments. We have made revisions to the resolution and fonts of the figures.
4. Include a critical comparison with at least three recent papers
Response: We are grateful for the constructive criticism that has been provided. We have elaborated in the second paragraph of the Introduction section.
5. Add a concise paragraph explicitly discussing the study’s limitations
Response: We are grateful for your constructive feedback. We have discussed the limitations of the manuscript in the sixth paragraph of the Discussion section.
Lines 360-375: The present study is subject to several limitations. First, the present study was conducted retrospectively and in a single-center study. Second, this study exclusively included older adult patients with osteoporotic hip fractures. Consequently, the findings may not be indicative of recurrent hip fractures in patients with traumatic or pathological fractures. Moreover, the study did not document the bone mineral density or biomarkers of the patients, primarily due to the absence of regular monitoring. Consequently, prospective studies will be designed to explore the inclusion of frailty indices or biomarkers, with a view to improving the accuracy of the models. Ultimately, the patients' functional baseline, that is to say, their mobility and capacity for activities of daily living, remained unassessed. Notwithstanding, the study provides sufficient data to identify the risk factors for recurrent hip fractures following surgical treatment of primary osteoporotic hip fractures, thus providing a valuable foundation for the clinical practice of surgeons. The integration of this model into electronic medical records is envisaged as a future development, with the objective of facilitating automated risk scoring. The delivery of targeted interventions to high-risk older patients is to be accomplished through the conduction of dual-energy X-ray absorptiometry (DXA) screenings, the initiation of osteoporosis treatment, and the implementation of fall prevention programs.
6. The paper is generally well-written but would benefit from minor grammatical polishing and simplification of long sentences.
Response: We appreciate your valuable comments. We have simplified the long sentences in the manuscript and corrected the minor grammatical errors.
7. Expand the conclusion to outline potential translational applications, such as clinical diagnostics, public health policy integration, or biomarker validation studies.
Response: We are grateful for the constructive criticism that has been provided. We have added the content in the Conclusion section of the manuscript and marked it in red font.
Lines 382-383: Therefore, the present study may also provide a theoretical foundation for clinical diagnostics, the integration of public health policy, and the validation of biomarkers.

Reviewer 4 Report
Comments and Suggestions for Authors
The scientific paper "Risk Factors for Recurrent hip Fractures Following Surgical Treatment of Primary Osteoporotic Hip Fractures in Elderly" aimed to analyze the risk factors for recurrent hip fractures following surgical treatment of primary osteoporotic hip fracture in elderly. The authors concluded that elderly patients with advanced age, COPD and PD were at greater risk of recurrent hip fractures. Early nutrition intervention and antiosteoporosis drug therapy may reduce the incidence of recurrent hip fractures in elderly following primary osteoporotic hip fracture surgery. After careful reading, I can make the following considerations:
1) Check the similarity index using the anti-plagiarism program, which is at 31% (iThenticate report), well above the acceptable level.
2) Abbreviations used in the abstract must contain their full meaning in the first citation.
3) The letter p (statistics) must be lowercase and italicized.
4) Insert reference on lines 47 and 62 of the introduction (end of paragraph).
5) In the methodology, include ethical aspects.
6) In the footer of the tables, insert the meaning of the abbreviations used.
7) On line 222, there is a citation of a supplementary table, not attached in the system.
8) In conclusion, include a broader contextualization of the study, its clinical applicability and innovation.
Overall, the research is interesting and has potential for publication, but needs to be improved.
Author Response
Comments and Suggestions for Authors
The scientific paper "Risk Factors for Recurrent hip Fractures Following Surgical Treatment of Primary Osteoporotic Hip Fractures in Elderly" aimed to analyze the risk factors for recurrent hip fractures following surgical treatment of primary osteoporotic hip fracture in elderly. The authors concluded that elderly patients with advanced age, COPD and PD were at greater risk of recurrent hip fractures. Early nutrition intervention and antiosteoporosis drug therapy may reduce the incidence of recurrent hip fractures in elderly following primary osteoporotic hip fracture surgery. After careful reading, I can make the following considerations:
- Check the similarity index using the anti-plagiarism program, which is at 31% (iThenticate report), well above the acceptable level.
Response: We are grateful for the constructive criticism that has been provided. We have made revisions to the contend of the manuscript. The similarity index is likely to be around 18%.
- Abbreviations used in the abstract must contain their full meaning in the first citation.
Response: We appreciate your valuable comments.
Lines 384-399:We have listed all the abbreviations. Newly added abbreviations are marked in red font.
- The letter p (statistics) must be lowercase and italicized.
Response: We are grateful for your constructive feedback. The letter p (statistics) has been revised.
- Insert reference on lines 47 and 62 of the introduction (end of paragraph).
Response: We appreciate your valuable suggestions. We have inserted reference on lines 47 the introduction of original manuscript (end of paragraph). Line 62 of the original manuscript is a summary statement, so no reference insertion is required.
Lines 61-63: It is widely established that the presence of a primary fragility fracture is frequently indicative of osteoporosis, and thus rendering the patient a risk factor for subsequent fractures [7].
- In the methodology, include ethical aspects.
Response: We appreciate your valuable comments. The content for the ethical section in the Methods has been added.
Lines 102-103: The present investigation was granted approval from the Medical Ethics Committee of Zhongshan Hospital, Fudan University (B2024-479R).
- In the footer of the tables, insert the meaning of the abbreviations used.
Response: We appreciate your valuable suggestions. We have added the meanings of the abbreviations below the tables.
Lines 217-219: Abbreviations: BMI, body mass index; ASA, American society of anesthesiologists; COPD, chronic obstructive pulmonary disease; PD, Parkinson’s disease; Hb, hemoglobin; eGFR, estimated Glomerular Filtration Rate; IQR, interquartile range; SD, standard deviation.
Lines 223-224: Abbreviations: COPD, chronic obstructive pulmonary disease; PD, Parkinson’s disease; B, beta; OR, odd ratios; CI, confidence interval.
Line 280: Abbreviations: B, beta; OR, odd ratios; CI, confidence interval.
- On line 222, there is a citation of a supplementary table, not attached in the system.
Response: We are grateful for your constructive feedback. We will re-upload an independent supplementary table during the revision process.
- In conclusion, include a broader contextualization of the study, its clinical applicability and innovation.
Response: We appreciate your valuable comments. We have added this content in the Conclusion section of the manuscript and marked it in red font.
Lines 382-383: Therefore, the present study may also provide a theoretical foundation for clinical diagnostics, the integration of public health policy, and the validation of biomarkers.
- Overall, the research is interesting and has potential for publication, but needs to be improved.
Response: We sincerely appreciate your recognition of our work and the favorable assessment provided in your review.

Reviewer 5 Report
Comments and Suggestions for Authors
The study examines the risk factors for hip fractures after surgical treatment in elderly.
Major concerns and comments:
- Table 1: Would "age+gender" together makes any significant difference?
- Table 1: what is ASA?
- Table 1: Any impact of different length of alcohol use?
- Table 1: Any impact of different length of COPD?
- Any impact of different length of dementia?
- Table 1: what is PD?
- Table 1: any effect of HbA1c? vitamin D level?
- Table 2: what is "B"?
- Figure 3: please add standard deviation or error on each bar.
- Please list all abbreviations.
- The patients were from China, therefore, it would be better to add "in Elderly in China" in title.
Author Response
Comments and Suggestions for Authors
The study examines the risk factors for hip fractures after surgical treatment in elderly.
Major concerns and comments:
1.Table 1: Would "age+gender" together makes any significant difference?
Response: We appreciate your valuable comments. We incorporated gender and age into the regression analysis. The results of the regression analysis revealed that advanced age was a risk factor for recurrent hip fractures, while gender did not enter the regression equation.
2.Table 1: what is ASA?
Response: Thanks for your question. ASA, American society of anesthesiologists. The ASA grade is a standard for assessing a patient's physical status that was developed by the American Society of Anesthesiologists and is most widely used in the field of anesthesia worldwide. Its core purpose is to quickly evaluate a patient's overall preoperative health status through a simple classification system, predict the risks of anesthesia and surgery, and provide important references for the formulation of anesthesia plans, the selection of surgical timing, and the allocation of medical resources.
Lines 384-399:We have listed all the abbreviations.
3.Table 1: Any impact of different length of alcohol use?
Response: We are indebted to you for your question. Since this study is a retrospective study, there was no specific information on alcohol consumption duration in the electronic medical records, thus this part of the content was not included in the analysis. We can conduct further investigation and analysis through prospective studies in the future.
4.Table 1: Any impact of different length of COPD?
Response: Your query is greatly appreciated. Since this study is a retrospective study, the information on comorbidities recorded in the electronic medical records was not complete, and thus the duration of comorbidities was not included in the analysis. We can conduct further investigation and analysis through prospective studies in the future.
5.Any impact of different length of dementia?
Response: Your enquiry has been received and is greatly appreciated. Since this study is a retrospective study, the information on comorbidities recorded in the electronic medical records was not complete, and thus the duration of comorbidities was not included in the analysis. We can conduct further investigation and analysis through prospective studies in the future.
6.Table 1: what is PD?
Response: Thanks for your question. PD, Parkinson’s disease.
Lines 384-399:We have listed all the abbreviations.
7.Table 1: any effect of HbA1c? vitamin D level?
Response: We are indebted to you for your question. Since not every patient underwent testing for glycated hemoglobin (HbA1c), this part of the data was missing and thus was not included in this study. This study found that there was no significant difference in the level of 25-hydroxyvitamin D between the recurrent hip fracture group and the non-recurrent hip fracture group.
8.Table 2: what is "B"?
Response: Thanks for your question. In regression analysis, "B" refers to the regression coefficient (Beta). The B value is a core indicator that measures the strength and direction of the linear relationship between an independent variable and a dependent variable.
9.Figure 3: please add standard deviation or error on each bar.
Response: We appreciate your valuable comments. Figure 3 (bar chart) depicts the cumulative mortality rate between the recurrent hip fracture group and non-recurrent hip fracture group. Therefore, no standard deviation (SD) needs to be added. Since it is a cumulative measure that counts the total number of events up to a certain point in time, and does not require the calculation of a standard deviation.
10.Please list all abbreviations.
Response: We appreciate your valuable suggestions. All abbreviations have been listed.
Lines 384-399:We have listed all the abbreviations. Newly added abbreviations are marked in red font.
11.The patients were from China, therefore, it would be better to add "in Elderly in China" in title.
Response: Your comments are greatly appreciated.
Line 2: We have revised the title of the manuscript to “Risk Factors for Recurrent Hip Fractures Following Surgical Treatment of Primary Osteoporotic Hip Fractures in Chinese Older Adults”.

Round 2
Reviewer 1 Report
Comments and Suggestions for Authors
The authors have responded thoroughly all my reviewer concerns
Reviewer 3 Report
Comments and Suggestions for Authors
Authors have addressed all the comments raised and now can be accepted for publication in its current form.
Reviewer 4 Report
Comments and Suggestions for Authors
I thank the authors for making the changes I suggested.
Reviewer 5 Report
Comments and Suggestions for Authors
No more comments.